# A Spectroscopic Validation of the Improved Lennard–Jones Model

**DOI:** 10.3390/molecules26133906

**Published:** 2021-06-26

**Authors:** Rhuiago Mendes de Oliveira, Luiz Guilherme Machado de Macedo, Thiago Ferreira da Cunha, Fernando Pirani, Ricardo Gargano

**Affiliations:** 1Instituto Federal de Educação, Ciência e Tecnologia do Maranhão, Bacabal, MA 65700-000, Brazil; rhuiago.oliveira@ifma.edu.br; 2Campus Centro-Oeste Dona Lindu—CCO, Universidade Federal de São João del Rei, Divinópolis, MG 35501-296, Brazil; lgm@ufsj.edu.br; 3Faculdade Evangélica de Goianésia, Unievangélica, Goianésia, GO 75083-515, Brazil; thiagofc88@hotmail.com; 4Dipartimento di Chimica, Biologia e Biotecnologie, Universitá degli studi di Perugia, via Elce di Sotto 8, 06123 Perugia, Italy; pirani.fernando@gmail.com; 5Istituto CNR di Scienze e Tecnologie Chimiche (CNR-SCITEC), via Elce di Sotto 8, 06123 Perugia, Italy; 6Instituto de Física, Universidade de Brasília, Brasília, DF 70910-900, Brazil

**Keywords:** noble gas molecules, rovibrational energies, lifetime, spectroscopic constants, improved Lennard Jones model

## Abstract

The Lennard–Jones (LJ) and Improved Lennard–Jones (ILJ) potential models have been deeply tested on the most accurate CCSD(T)/CBS electronic energies calculated for some weakly bound prototype systems. These results are important to plan the correct application of such models to systems at increasing complexity. CCSD(T)/CBS ground state electronic energies were determined for 21 diatomic systems composed by the combination of the noble gas atoms. These potentials were employed to calculate the rovibrational spectroscopic constants, and the results show that for 20 of the 21 pairs the ILJ predictions agree more effectively with the experimental data than those of the LJ model. The CCSD(T)/CBS energies were also used to determine the β parameter of the ILJ form, related to the softness/hardness of the interacting partners and controlling the shape of the potential well. This information supports the experimental finding that suggests the adoption of β≈9 for most of the systems involving noble gas atoms. The He-Ne and He-Ar molecules have a lifetime of less than 1ps in the 200–500 K temperature range, indicating that they are not considered stable under thermal conditions of gaseous bulks. Furthermore, the controversy concerning the presence of a “virtual” or a “real” vibrational state in the He${}_{2}$ molecule is discussed.

## 1. Introduction

The detailed characterization of several equilibrium and non-equilibrium properties of matter (in condensed and gaseous phases) is often obtained through the proper formulation of force fields associated with non-covalent intermolecular interactions [1]. The adoption of simple and accurate models of this type of interactions, to be easily used in molecular dynamics simulations of both ionic and neutral aggregates, still represents a basic question. In particular, such models must be given in the analytical form, from which the first and second derivative of the interaction, defining force, and force constant must be easily obtained and must present continuity of behavior. Moreover, they must involve few parameters having a defined physical meaning that can be used as proper scaling factors when the extension to systems at increasing complexity is attempted. This target can be achieved by investigating in detail prototype systems for which accurate experimental and theoretical information on the intermolecular interaction is easily obtainable.

The venerable Lennard–Jones (LJ) analytical form [2] is still widely used in the molecular dynamics simulations of systems dominated by van der Waals interactions. This model meets some of the requests stressed above but shows some important deficiencies, especially at large and short intermolecular distances. Several intermolecular pair potential formulations have been proposed to overcome these limitations [3,4,5,6,7,8,9,10,11,12,13,14,15,16,17,18,19]. In general, these alternative models use a combination of complicated functions and with many adjustable parameters.

Some years ago the adoption of an Improved Lennard–Jones (ILJ) function [20] permitted to obtain, for many noble gas pairs, the most accurate experimental values of well depth D${}_{e}$ and equilibrium distance R${}_{e}$ from the combined analysis of scattering experiments, with the resolution of fundamental quantum interference effects, spectroscopy, and transport properties. In the same paper, it has been also demonstrated that while ILJ provides asymptotically a dipole–dipole dispersion attraction coefficient equal to $C_{6}=D_{e}\cdot R_{e}^{6}$, which is in good agreement with the most accurate theoretical and experimental values, the LJ model predicts $C_{6}=2D_{e}\cdot R_{e}^{6}$ a factor 2 larger, with also a poor reproduction of the experimental observables. Moreover, values in the range of 7 to 9 (depending on the softness of the interacting partners) of the additional parameter β in ILJ formulation (See next section) work well for several neutral-neutral and ion-neutral cases [20,21]. Such values allowed a proper assessment of the role of the van der Waals interaction component in the formation of the weak hydrogen and halogen intermolecular bonds [21,22]. The excessive long-range attraction of LJ can be a strong limitation when the model is applied, as often made, to describe the behavior of big molecules, where many interaction centers are involved, several of them separated by large distances. For the application of ILJ function to systems at increasing complexity, like those involving biomolecules, the selective passage of chemical species in cellular channels and pores, the physical adsorption on single and multiple layers, further tests with a possible generalization of its formulation are desirable and probably necessary. The achievement of this target can be pursued through a sequence of steps.

In this paper, we test in detail the shape of the potential well predicted by ILJ and LJ on accurate ab initio values of the interaction and the combined analysis of spectroscopic features of both symmetric and asymmetric noble gas dimers. This study confirms that ILJ with β≈9 provides the best representation for most of the investigated systems. The following steps should involve an accurate analysis of short-range repulsion of atom–atom systems with great difference in the polarizability (or in softness) to test the modulation of the repulsion by varying the β value. Moreover, other important information can be provided by a further accurate study of systems, formed by neutral partners interacting with both negative and single or multiple charged positive ions. For such systems, the depth, location, and shape of the potential well and the steepness of the first part of the repulsion arise from a more critical balance of the attraction, stronger respect to the neutral-neutral cases, and of the repulsion.

This work is organized as follows. The methodologies adopted in this study are summarized in Section 2, while obtained results and discussion are presented in Section 3, and, finally, some conclusions are provided in Section 4.

## 2. Methodologies

The ground state potential energy curves (PECs) of 21 diatomic molecules, weakly bound by prototype non-covalent forces, were determined by combining the CCSD(T) [23,24] and basis set superposition error correction [25,26] methods. Associated with both methods, it was used the aug-cc-pVQZ, aug-cc-pV5Z basis sets [27,28,29], and complete basis set (CBS) approach [30,31,32].

To correctly describe the electronic structure of heavier elements, it is necessary to include the relativistic effects [33,34,35]. As examples, (i) relativistic effects account for 1.7–1.8 V in a standard 2 V lead-acid battery cell [36], and (ii) the non-relativistic gold is white (like silver) so the yellow color of gold comes from relativity [34]. A typical way of including relativity in electronic structure calculations is through the use of pseudopotentials [37,38] (PP). The relativistic pseudopotentials used in this investigation for Xe and Rn were the small core energy consistent developed by Peterson et al. [39], which were adjusted to multiconfiguration Dirac–Hartree–Fock data based on Dirac–Coulomb–Breit Hamiltonian, with the accompanying aug-cc-pVQZ-PP and aug-cc-pV5Z-PP basis sets. Note that errors due to the used pseudopotentials are found negligible [39] (they are expected to provide a maximum contribution to D${}_{e}$ of about 1.3 KJ/mol). In addition, the selected PP with the matched basis sets exhibits the systematic convergence and accuracy characteristic of their all-electron counterparts also used in this investigation for lighter He, Ne, Ar, and Kr [28,29]. All these PECs were calculated through the Gaussina09 computational code [40].

Through the $R_{e}$ and $D_{e}$ calculated values, combined with an accurate investigation of the radial dependence of the interaction, effective isotropic PECs have been constructed exploiting the LJ and ILJ analytical forms. The general formulation of the classical LJ model is given by the following equation:(1)$$V\left(R\right)=D_{e}\left[\frac{m}{n-m}{\left(\frac{R_{e}}{R}\right)}^{n}-\frac{n}{n-m}{\left(\frac{R_{e}}{R}\right)}^{m}\right]$$
that for neutral-neutral systems, with n=12 and m=6, this equation turns into the following well-known form:(2)$$V\left(R\right)=D_{e}\left[{\left(\frac{R_{e}}{R}\right)}^{12}-2{\left(\frac{R_{e}}{R}\right)}^{6}\right]$$

For the ILJ function it has been proposed that
(3)$$V\left(R\right)=D_{e}\left[\frac{m}{n(R)-m}{\left(\frac{R_{e}}{R}\right)}^{n(R)}-\frac{n(R)}{n(R)-m}{\left(\frac{R_{e}}{R}\right)}^{m}\right]$$
where $n\left(R\right)=\beta +4{\left(\frac{R}{R_{e}}\right)}^{2}$ and β parameter describes the softness/hardness of the elements involved in the complex and β is experimentally set to 9 for systems involving noble gases [20]. For neutral-neutral systems, *m* assumes the value of 6 and the ILJ form becomes
(4)$$V\left(R\right)=D_{e}\left[\frac{6}{n(R)-6}{\left(\frac{R_{e}}{R}\right)}^{n(R)}-\frac{n(R)}{n(R)-6}{\left(\frac{R_{e}}{R}\right)}^{6}\right]$$

Rovibrational energies of each diatomic molecule were determined by solving the nuclear Schrödinger equation. To solve this equation, the Discrete Variable Representation method [41] was employed. Rovibrational spectroscopic constants, such as $\omega_{e}$, $\omega_{e}x_{e}$, $\omega_{e}y_{e}$, $\alpha_{e}$, and $\gamma_{e}$, were calculated using the following expressions [42]:(5)$$\begin{array}{cc} \omega_{e} & =\frac{1}{24}\left[141\left(E_{1,0}-E_{0,0}\right)-93\left(E_{2,0}-E_{0,0}\right)+23\left(E_{3,0}-E_{1,0}\right)\right]\\ \omega_{e}x_{e} & =\frac{1}{4}\left[13\left(E_{1,0}-E_{0,0}\right)-11\left(E_{2,0}-E_{0,0}\right)+3\left(E_{3,0}-E_{1,0}\right)\right]\\ \omega_{e}y_{e} & =\frac{1}{6}\left[3\left(E_{1,0}-E_{0,0}\right)-3\left(E_{2,0}-E_{0,0}\right)+\left(E_{3,0}-E_{1,0}\right)\right]\\ \alpha_{e} & =\frac{1}{8}\left[-12\left(E_{1,1}-E_{0,1}\right)+4\left(E_{2,1}-E_{0,1}\right)+4\omega_{e}-23\omega_{e}y_{e}\right]\\ \gamma_{e} & =\frac{1}{4}\left[-2\left(E_{1,1}-E_{0,1}\right)+\left(E_{2,1}-E_{0,1}\right)+2\omega_{e}x_{e}-9\omega_{e}y_{e}\right] \end{array}$$

In Equation (Equation 5), $E_{v,j}$ represents the rovibrational energy, where the indices *v* and *j* indicate the vibrational and rotational quantum numbers, respectively. To verify the accuracy of spectroscopic constants, the Dunham method [43] was also used. This approach depends on the derivatives of PECs in the equilibrium configuration.

For each Ng-Ng molecules, the lifetime as a function of temperature was calculated using Slater’s method which is described by the equation [44,45]:(6)$$\tau \left(T\right)=\frac{1}{\omega_{e}}e^{\frac{D_{e}-E_{0,0}}{RT}}$$

In the Equation (Equation 6), *T* is the temperature, *R* the universal gas constant, and $E_{0,0}$ the zero-point energy. This equation provides the lifetime for decomposition of the systems and it is a description purely dynamical with a vibrational analysis of the complexes, referring to the low or high rate of unimolecular decay and it is supposed to occur when the interaction coordinate reaches the dissociation threshold ($D_{e}$). In general, this approach is suitable for regions of intermediate pressure in the bulk.

## 3. Results and Discussions

### 3.1. Equilibrium Distances, Dissociation Energies and Potential Energy Curves

Table 1 shows the CCSD(T)/aug-cc-pVQZ, CCSD(T)/aug-cc-pV5Z, CCSD(T)/CBS, and experimental equilibrium distances for all Ng-Ng diatomic molecules (with Ng = He, Ne, Ar, Kr, Xe, and Rn). From this table it is possible to note that the equilibrium distances calculated with CCSD(T)/CBS level agree more effectively with experimental data [20,46]. CCSD(T)/aug-cc-pVQZ, CCSD(T)/aug-cc-pV5Z, CCSD(T)/CBS results and experimental dissociation energies, for the Ng-Ng molecules, are compared in Table 2. These results also show that the best agreement between theoretical and experimental data happens with the CCSD(T)/CBS level, mainly when compared with the data available in [20,47].

The twenty-one complete PECs for all systems with CCSD(T)/aug-cc-pVQZ, CCSD(T) /aug-cc-pV5Z, and CCSD(T)/CBS levels are shown in the d, Appendix A (from Appendix A). These PECs were built calculating the ground state electronic energies for different values of the internuclear distances (R) that ranged from the region of the strong interaction (R less than R${}_{e}$) to the asymptotic region (R much larger than R${}_{e}$). For R less and greater than equilibrium distance (R${}_{e}$), it was used a step of 0.1 Å, while for R near to R${}_{e}$ was considered a step of 0.01 Å. With these steps, it was determined approximately a hundred electronic energies for all Ng-Ng molecules (except for the Kr-Rn, Xe-Rn, and Rn${}_{2}$ systems).

The β parameter of ILJ model (Equation (Equation 4)) was determined for each molecule by fitting, via Powell method [48], the set of CCSD(T)/CBS electronic energies as shown in Table 3. From this table, it is possible to note that β parameter, which describes the softness/hardness of the elements involved in the molecule, for each molecule is very close to the experimental value. This fact supports the experimental prediction that this parameter is close to 9 for most molecules formed with noble gases. The root means square deviation of the performed fitting varied from 3.97 × 10${}^{-5}$ Hartree (for Ar-Kr system) to 1.05 × 10${}^{-7}$ Hartree (for He${}_{2}$ system) for all considered molecules (see Appendix A of Appendix A).

### 3.2. Rovibrational Energies, Spectroscopic Constants, and Lifetime

Once the CCSD(T)/CBS ILJ PEC of the 21 studied molecules were obtained, their rovibrational energies were calculated using the reduced mass showed in Appendix A of Appendix A and they can be found in Appendix A of Appendix A. The experimental vibrational energy spacings for the Ne${}_{2}$ (1 transition), Ar${}_{2}$ (5 transitions), Kr${}_{2}$ (9 transitions), and Xe${}_{2}$ (10 transitions) systems [20] were compared with the present results. From this comparison, it was found a difference of 0.38 cm${}^{-1}$ for Ne${}_{2}$ (1–0 transition) and a maximum and minimum difference of 0.47 cm${}^{-1}$ (1–0 transition) and 0.09 cm${}^{-1}$ (2–1 transition) for Ar${}_{2}$, 0.60 cm${}^{-1}$ (1–0 transition) and 0.00 cm${}^{-1}$ (8–7 transition) for Kr${}_{2}$, and 0.27 cm${}^{-1}$ (3–2 transition) and 0.01 cm${}^{-1}$ (8–7 transition) for Xe${}_{2}$, respectively. Furthermore, from the point of view of the CCSD(T)/CBS calculation (D${}_{e}=0.914$ meV, R${}_{e}=2.97$Å, and β=8.68), the He${}_{2}$ system does not present a vibrational level within the PEC, i.e., the He${}_{2}$ first rovibrational state is considered virtual (unbound). This fact agrees with that found by Wang et al. [49], but disagrees with the calculations conducted by Aziz et al. [50] (based on the LM2M2 semiempirical potential) and Tang et al. [51] that predict one weakly bound state for ${}^{4}$H$e_{2}$ dimer. This controversy is expected because ${}^{4}$H$e_{2}$ dimer interactions are composed by the combination of small mass and small atomic polarizability and it makes that the rovibrational energy of the lowest state places very close to that of the separated atoms. In particular, the small potential well arises from the critical combination of a limited repulsion with the weakest attraction existing in nature and it controls many peculiarities of He in gaseous [52] and condensed phases, as its anomalous phase diagram. This issue was resolved experimentally by Luo et al. [53,54] with the mass spectrometric observation of bound ${}^{4}$H$e_{2}$ in an extreme pulsed supersonic beam of He at temperatures less than 1mK. This fact was confirmed by Schöllkopf et al. [55] by using a novel diffraction experiment employing a nanoscale transmission grating. Finally, the current results suggest that, in addition to the He${}_{2}$ system, also the He-Ne, He-Ar, He-Kr, and He-Rn molecules are less stable, as they only showed two vibrational levels within their PECs.

Moreover, to emphasize the ILJ sensitivity to the potential parameters and to cast further light on the controversy concerning the presence of a “virtual” or a “real” rovibrational state in He${}_{2}$ molecule, confined at the dissociation limit of a very small potential well, we modulated slightly shape and depth of the potential well of the ILJ potential formulation. In particular, the shape has been adjusted by lowering β (maintaining always its value within the limit 7–9 typical of van der Waals forces for neutral-neutral systems [20,21] and accompanying it by a maximum D${}_{e}$ increase of 0.1 meV (0.01 KJ/mol) respect to the CCSD(T)/CBS result, to include in this increase any possible uncertainty of ab initio calculations. The new ILJ formulation, adopting β=7.6, D${}_{e}$ = 0.988 meV (7.9687 cm${}^{-1}$) and R${}_{e}$ = 2.974 Å, provides results still consistent with the experimental determination [46] and with those of other more sophisticated potential models [51,52] in an extended range of 2.0 to 6.0 Å of internuclear distances, and its well contains here a “real” vibrational state with a value of 7.9685 cm${}^{-1}$.

To verify the influence of the $R_{e}$, $D_{e}$, and β parameters on the quality of the ILJ PEC, the rovibrational spectroscopic constants (RSC) of the 21 molecules were calculated considering these parameters determined at CCSD(T)/CBS level and with an experimental value of β. Table 4, Table 5 and Table 6 show the RSC calculated by using both Equation (Equation 5) (whose rovibrational energies were calculated using the DVR method) and Dunham method. It is important to mention that Equation (Equation 5) can only be used for systems that have at least four vibrational levels within the PEC well. Thus, the RSC for the He-Ne, He-Ar, He-Kr, He-Xe, He-Rn, and Ne-Ne molecules were only determined by the Dunham method. From these Tables, note that the RSC determined with ILJ model agrees more with the experimental data than LJ representation for twenty of the twenty-one studied molecules (except the He-Ne molecule). For almost all of the 21 studied molecules (except for He-Ar, He-Kr, He-Ne, Ne-Xe, and Ar-Kr systems), the RSC agrees more with experimental data when an ILJ PEC with β=9 (experimental value) is used. This fact suggests that β=9 is an accurate choice to describe molecular systems involving noble gases.

To specify each type of calculation, the following nomenclatures were used: D-ILJ-β9 (RSC calculated with Dunham method and an ILJ PEC with R${}_{e}$, D${}_{e}$, and β given by experimental values), D-ILJ-CBS-β9 (RSC calculated with Dunham method and a PEC ILJ with R${}_{e}$ and D${}_{e}$ obtained at CCSD(T)/CBS level and with an experimental value of β), D-ILJ-CBS-βFIT (RSC calculated with Dunham method and a PEC ILJ with R${}_{e}$ and D${}_{e}$ determined at CCSD(T)/CBS level and β fitted from CCSD(T)/CBS energies), DVR-ILJ-CBS-β9 (RSC calculated with DVR method and a PEC ILJ with R${}_{e}$ and D${}_{e}$ obtained at CCSD(T)/CBS level and with a experimental value of β), DVR-ILJ-CBS-βFIT (RSC calculated with DVR method and a PEC ILJ with R${}_{e}$ and D${}_{e}$ obtained at CCSD(T)/CBS level and β fitted from CCSD(T)/CBS energies), D-LJ-CBS (RSC calculated with Dunham method and a PEC LJ with R${}_{e}$ and D${}_{e}$ determined at CCSD(T)/CBS level), and DVR-LJ-CBS (RSC calculated with DVR method and a PEC LJ with R${}_{e}$ and D${}_{e}$ determined at CCSD(T)/CBS level). For the He-Xe system, the RSC were calculated using the β parameter obtained at aug-cc-pV5Z basis set because the β adjustment for the CBS base did not converge and it is indicated in the Table 5 by the D-ILJ-5z-βFIT symbol.

Figure 1 shows the lifetime as a function of temperature for all studied molecules, except for the He${}_{2}$ system that has no vibrational level within its PEC well. From this figure, one can see that He-Ne and He-Ar molecules have a lifetime of over 1.0 picosecond for all considered temperature ranges (200–500 K) and that the He-Kr lifetime is slightly larger than 1 picosecond within the same temperature range. Following the recommendations of wolfgang [56], which states that a lifetime over 1.0 picosecond means that the PEC well is not deep enough to exclude the intermediate complex, these compounds can be considered unstable. It is not possible to determine the He${}_{2}$ lifetime, because no vibrational level or only one bound state at the dissociation limit was found within the He${}_{2}$ PEC.

## 4. Conclusions

In this work, an accurate test of the β parameter value, defining the strength of both attraction and repulsion in the ILJ model (See Equations (3) and (4)), was obtained exploiting CCSD(T)/CBS electronic energies calculated for the complete family of the diatomic molecules formed by the He, Ne, Ar, Kr, Xe, and Rn noble gas atoms. For all considered molecules, β was found to be close to 9 and this feature is supported by both theoretical and experimental findings. To verify the influence of the shape of the potential well on the observables, ILJ PEC has been adopted to predict rovibrational energies, spectroscopic constants, and lifetime as a function of temperatures. The results suggest that ILJ analytical form with β≈9 provides rovibrational spectroscopic constants (RSC) that agree more effectively with experimental than RSC determined with LJ PEC. This fact confirms that most of the LJ inadequacies at large and short intermolecular distances are overcome by the ILJ model. Predicted lifetimes indicate that the He-Ne and He-Ar molecules are not stable under temperature confined in the 200 to 500 K range.

We found that an increase in He${}_{2}$ well depth of less than 0.1 meV (0.01 KJ/mol) accompanied by a slight change in its shape and position of the well (some fraction of a hundredth of Angstrom), all the characteristics that arise from a very critical balance of weak attraction with the repulsion, leads to the existence of a real vibrational level. Note that these changes are within the errors of any ab initio calculation, even of the CCSD(T)/CBS type that extends to long-range asymptotic regions.

Finally, as an important conclusion, further investigations, carried out combining accurate theoretical and experimental information and focused on the critical balance of attraction and repulsion controlled by the n(R) term, are expected to be crucial for the correct modulation of β parameter and of the numerical coefficient 4 (See Equations (3) and (4)) when systems with completely different nature and size are involved in non-covalent interactions.

## Figures and Tables

**Figure 1 molecules-26-03906-f001:**
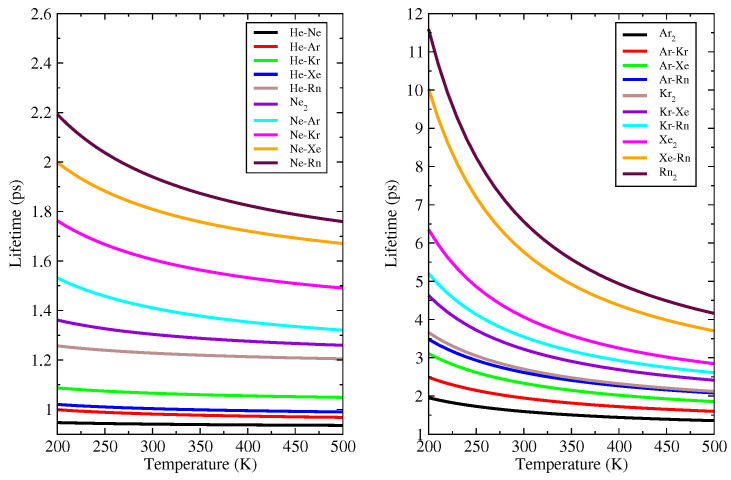
He-Ne, He-Ar, He-Kr, He-Xe, He-Rn, Ne${}_{2}$, Ne-Ar, Ne-Kr, Ne-Xe, Ne-Rn, Ar${}_{2}$, Ar-Kr, Ar-Xe, Ar-Rn, Kr${}_{2}$, Kr-Xe, Kr-Rn, Xe${}_{2}$, Xe-Rn e Rn${}_{2}$ lifetimes as a function of the temperature in the range between 200 K and 500 K.

**Table 1 molecules-26-03906-t001:** CCSD(T)/aug-cc-pVQZ, CCSD(T)/aug-cc-pV5Z, CCSD(T)/CBS, and experimental equilibrium distances (Å) for the Ng-Ng molecules (Ng = He, Ne, Ar, Kr, Xe, and Rn).

Molecules	aug-cc-pVQZ	aug-cc-pV5Z	CBS	Exp. [46]	Exp. [20]	Exp. [47]
He${}_{2}$	3.01	2.99	2.97	2.97	–	–
He-Ne	3.07	3.05	3.01	3.03	–	–
He-Ar	3.54	3.51	3.49	3.48	–	3.48
He-Kr	3.75	3.72	3.70	3.69	–	3.70
He-Xe	4.04	4.01	4.00	3.98	3.99	4.00
He-Rn	4.16	4.13	4.10	–	–	–
Ne${}_{2}$	3.15	3.13	3.10	3.09	3.09	–
Ne-Ar	3.55	3.52	3.48	3.49	3.52	–
Ne-Kr	3.72	3.69	3.65	3.62	3.66	–
Ne-Xe	3.96	3.9	3.90	3.86	3.88	–
Ne-Rn	4.06	4.02	3.98	–	–	–
Ar${}_{2}$	3.83	3.80	3.75	3.76	3.76	–
Ar-Kr	3.96	3.94	3.90	3.88	3.91	–
Ar-Xe	4.16	4.13	4.11	4.07	4.10	–
Ar-Rn	4.23	4.20	4.16	–	–	–
Kr${}_{2}$	4.09	4.06	4.04	4.01	4.01	–
Kr-Xe	4.27	4.25	4.22	4.17	4.20	–
Kr-Rn	4.34	4.31	4.27	–	–	–
Xe${}_{2}$	4.44	4.41	4.38	4.36	4.35	–
Xe-Rn	4.49	4.46	4.43	–	–	–
Rn${}_{2}$	4.54	4.50	4.47	–	–	–

**Table 2 molecules-26-03906-t002:** CCSD(T)/aug-cc-pVQZ, CCSD(T)/aug-cc-pV5Z, CCSD(T)/CBS, and experimental dissociation energies (meV) for the Ng-Ng molecules (Ng = He, Ne, Ar, Kr, Xe, and Rn).

Molecules	aug-cc-pVQZ	aug-cc-pV5Z	CBS	Exp. [46]	Exp. [20]	Exp. [47]
He${}_{2}$	0.806	0.849	0.914	0.944	–	–
He-Ne	1.522	1.655	1.893	1.782	–	–
He-Ar	2.204	2.373	2.599	2.492	–	2.59
He-Kr	2.254	2.429	2.638	2.478	–	2.67
He-Xe	2.172	2.349	2.551	2.356	2.624	2.64
He-Rn	2.138	2.323	2.536	–	–	–
Ne${}_{2}$	2.879	3.199	3.750	3.641	3.660	–
Ne-Ar	4.656	5.135	5.846	5.823	5.740	–
Ne-Kr	5.017	5.536	6.269	6.169	6.160	–
Ne-Xe	5.152	5.711	6.493	6.395	6.350	–
Ne-Rn	5.259	5.854	6.740	–	–	–
Ar${}_{2}$	10.295	11.239	12.357	12.343	12.370	–
Ar-Kr	12.119	13.165	14.373	15.658	14.330	–
Ar-Xe	13.770	14.993	16.410	16.253	16.090	–
Ar-Rn	14.758	16.100	17.632	–	–	–
Kr${}_{2}$	14.648	15.782	17.048	17.339	17.300	–
Kr-Xe	17.159	18.504	20.005	20.120	19.950	–
Kr-Rn	18.690	20.209	21.814	–	–	–
Xe${}_{2}$	20.878	22.501	24.307	24.327	24.200	–
Xe-Rn	23.220	25.019	26.970	–	–	–
Rn${}_{2}$	26.222	28.220	30.320	–	–	–

**Table 3 molecules-26-03906-t003:** β parameter values adjusted using the CCSD(T)/aug-cc-pVQZ, CCSD(T)/aug-cc-pV5Z, and CCSD(T)/CBS electronic energies for the Ng-Ng molecules (Ng = He, Ne, Ar, Kr, Xe, and Rn).

Molecules	aug-cc-pVQZ	aug-cc-pV5Z	CBS
He${}_{2}$	8.67	8.74	8.68
He-Ne	9.15	8.87	8.65
He-Ar	9.15	9.31	8.94
He-Kr	9.26	9.32	8.89
He-Xe	9.58	9.40	–
He-Rn	9.60	9.28	8.96
Ne${}_{2}$	9.70	9.18	8.33
Ne-Ar	9.40	9.34	9.23
Ne-Kr	9.63	9.49	9.27
Ne-Xe	9.78	9.46	8.71
Ne-Rn	9.49	9.31	8.85
Ar${}_{2}$	9.15	9.02	9.74
Ar-Kr	9.37	9.00	9.22
Ar-Xe	9.35	9.06	8.48
Ar-Rn	9.12	8.83	8.84
Kr-Kr	9.22	9.20	8.60
Kr-Xe	9.40	8.80	8.71
Kr-Rn	8.89	8.68	8.79
Xe${}_{2}$	9.24	9.03	8.74
Xe-Rn	9.17	8.75	8.40
Rn${}_{2}$	8.71	8.68	8.19

**Table 4 molecules-26-03906-t004:** He${}_{2}$, Ne${}_{2}$, Ar${}_{2}$, Kr${}_{2}$, Xe${}_{2}$, and Rn${}_{2}$ rovibrational spectroscopic constants (RSC) given in cm${}^{-1}$. The D-ILJ-β9 acronym stands for RSC calculated with Dunham method and an ILJ PEC with R${}_{e}$, D${}_{e}$, and β (equal 9) experimental values; The D-ILJ-CBS-β9 acronym stands for RSC calculated with Dunham method and a PEC ILJ with R${}_{e}$ and D${}_{e}$ obtained at CCSD(T)/CBS level and a β (equal 9) experimental value; The D-ILJ-CBS-βFIT acronym stands for RSC calculated with Dunham method and a PEC ILJ with R${}_{e}$ and D${}_{e}$ determined at CCSD(T)/CBS level and β fitted from CCSD(T)/CBS energies; The DVR-ILJ-CBS-β9 acronym stands for RSC calculated with DVR method and a PEC ILJ with R${}_{e}$ and D${}_{e}$ obtained at CCSD(T)/CBS level and a β (equal 9) experimental value; The DVR-ILJ-CBS-βFIT acronym stands for RSC calculated with DVR method and a PEC ILJ with R${}_{e}$ and D${}_{e}$ obtained at CCSD(T)/CBS level and β fitted from CCSD(T)/CBS energies; The D-LJ-CBS acronym stands for RSC calculated with Dunham method and a PEC LJ with R${}_{e}$ and D${}_{e}$ determined at CCSD(T)/CBS level; DVR-LJ-CBS acronym stands for RSC calculated with DVR method and a PEC LJ with R${}_{e}$ and D${}_{e}$ determined at CCSD(T)/CBS level.

Molecules	Methods	$\omega_{e}$	$\omega_{e}x_{e}$	$\omega_{e}y_{e}$	$\alpha_{e}$	$\gamma_{e}$
	Exp. [46]	33.2	–	–	–	–
	D-ILJ-β9	33.64	–	–	–	–
He${}_{2}$	D-ILJ-CBS-β9	33.13	–	–	–	–
	D-ILJ-CBS-βFIT	32.72	–	–	–	–
	D-LJ-CBS	31.83	–	–	–	–
	Exp. [46]	28.5	–	–	–	–
	D-ILJ-β9	28.35	7.75	1.8 × 10${}^{-1}$	3.6 × 10${}^{-2}$	4.0 × 10${}^{-3}$
Ne${}_{2}$	D-ILJ-CBS-β9	28.64	7.71	1.8 × 10${}^{-1}$	3.6 × 10${}^{-2}$	4.0 × 10${}^{-3}$
	D-ILJ-CBS-βFIT	27.89	7.16	1.1 × 10${}^{-1}$	3.5 × 10${}^{-2}$	3.0 × 10${}^{-3}$
	D-LJ-CBS	27.51	7.98	5.9 × 10${}^{-1}$	3.9 × 10${}^{-2}$	3.5 × 10${}^{-3}$
	Exp. [46]	30.9	–	–	–	–
	D-ILJ-β9	30.54	2.67	3.8 × 10${}^{-2}$	3.0 × 10${}^{-3}$	2.0 × 10${}^{-4}$
Ar${}_{2}$	DVR-ILJ-CBS-β9	30.60	2.69	3.8 × 10${}^{-2}$	3.9 × 10${}^{-3}$	2.4 × 10${}^{-4}$
	D-ILJ-CBS-β9	30.54	2.63	2.0 × 10${}^{-2}$	4.0 × 10${}^{-3}$	1.7 × 10${}^{-4}$
	DVR-ILJ-CBS-βFIT	31.48	2.91	5.2 × 10${}^{-2}$	4.0 × 10${}^{-3}$	2.5 × 10${}^{-4}$
	D-ILJ-CBS-βFIT	31.40	2.84	2.9 × 10${}^{-2}$	4.1 × 10${}^{-3}$	1.8 × 10${}^{-4}$
	DVR-LJ-CBS	29.37	2.75	7.5 × 10${}^{-2}$	4.3 × 10${}^{-3}$	1.5 × 10${}^{-4}$
	D-LJ-CBS	29.37	2.73	6.6 × 10${}^{-2}$	4.4 × 10${}^{-3}$	1.2 × 10${}^{-4}$
	Exp. [46]	23.6	–	–	–	–
	D-ILJ-β9	23.33	1.09	4.0 × 10${}^{-3}$	9.0 × 10${}^{-4}$	2.1 × 10${}^{-5}$
Kr${}_{2}$	DVR-ILJ-CBS-β9	22.99	1.08	6.1 × 10${}^{-3}$	9.0 × 10${}^{-4}$	2.5 × 10${}^{-5}$
	D-ILJ-CBS-β9	22.99	1.08	4.5 × 10${}^{-3}$	9.0 × 10${}^{-4}$	2.1 × 10${}^{-5}$
	DVR-ILJ-CBS-βFIT	22.64	1.04	4.9 × 10${}^{-3}$	8.9 × 10${}^{-4}$	2.4 × 10${}^{-5}$
	D-ILJ-CBS-βFIT	22.63	1.03	3.5 × 10${}^{-3}$	8.9 × 10${}^{-4}$	2.0 × 10${}^{-5}$
	DVR-LJ-CBS	22.09	1.12	1.5 × 10${}^{-2}$	9.8 × 10${}^{-4}$	1.7 × 10${}^{-5}$
	D-LJ-CBS	22.09	1.12	1.4 × 10${}^{-2}$	9.9 × 10${}^{-4}$	1.5 × 10${}^{-5}$
	Exp. [46]	20.9	–	–	–	–
	D-ILJ-β9	20.33	0.59	1.0 × 10${}^{-3}$	3.0 × 10${}^{-4}$	4.5 × 10${}^{-6}$
Xe${}_{2}$	DVR-ILJ-CBS-β9	20.24	0.58	1.8 × 10${}^{-3}$	3.0 × 10${}^{-4}$	4.9 × 10${}^{-6}$
	D-ILJ-CBS-β9	20.24	0.58	1.5 × 10${}^{-3}$	3.0 × 10${}^{-4}$	4.3 × 10${}^{-6}$
	DVR-ILJ-CBS-βFIT	20.03	0.57	1.5 × 10${}^{-3}$	3.0 × 10${}^{-4}$	4.8 × 10${}^{-6}$
	D-ILJ-CBS-βFIT	20.03	0.57	1.3 × 10${}^{-3}$	3.0 × 10${}^{-4}$	4.2 × 10${}^{-6}$
	DVR-LJ-CBS	18.24	0.59	5.3 × 10${}^{-3}$	3.3 × 10${}^{-4}$	3.5 × 10${}^{-6}$
	D-LJ-CBS	19.44	0.60	4.9 × 10${}^{-3}$	3.3 × 10${}^{-4}$	3.2 × 10${}^{-6}$
	DVR-ILJ-CBS-β9	16.36	0.32	6.9 × 10${}^{-4}$	1.1 × 10${}^{-4}$	1.2 × 10${}^{-6}$
Rn${}_{2}$	D-ILJ-CBS-β9	16.31	0.32	5.9 × 10${}^{-4}$	1.1 × 10${}^{-4}$	1.1 × 10${}^{-6}$
	DVR-ILJ-CBS-βFIT	15.87	0.30	4.1 × 10${}^{-4}$	1.1 × 10${}^{-4}$	1.2 × 10${}^{-6}$
	D-ILJ-CBS-βFIT	16.11	0.31	4.9 × 10${}^{-4}$	1.1 × 10${}^{-4}$	1.1 × 10${}^{-6}$
	DVR-LJ-CBS	15.67	0.34	1.9 × 10${}^{-3}$	1.2 × 10${}^{-4}$	9.0 × 10${}^{-7}$
	D-LJ-CBS	16.36	0.34	1.9 × 10${}^{-3}$	1.2 × 10${}^{-4}$	8.4 × 10${}^{-4}$

**Table 5 molecules-26-03906-t005:** He-Ne, He-Ar, He-Kr, He-Xe, He-Rn, Ne-Ar, Ne-Kr, and Ne-Xe rovibrational spectroscopic constants (RSC) given in cm${}^{-1}$.

Molecules	Methods	$\omega_{e}$	$\omega_{e}x_{e}$	$\omega_{e}y_{e}$	$\alpha_{e}$	$\gamma_{e}$
	Exp. [46]	35.0	–	–	–	–
	D-ILJ-β9	35.57	25.79	1.46	2.9 × 10${}^{-1}$	9.8 × 10${}^{-2}$
He-Ne	D-ILJ-CBS-β9	36.42	26.10	1.47	2.9 × 10${}^{-1}$	9.7 × 10${}^{-2}$
	D-ILJ-CBS-βFIT	35.93	25.10	1.19	2.9 × 10${}^{-1}$	9.0 × 10${}^{-2}$
	D-LJ-CBS	34.99	26.94	0.32	3.1 × 10${}^{-1}$	7.2 × 10${}^{-1}$
	Exp. [46]	34.8	–	–	–	–
	D-ILJ-β9	35.31	17.36	7.1 × 10${}^{-1}$	1.4 × 10${}^{-1}$	3.3 × 10${}^{-2}$
He-Ar	D-ILJ-CBS-β9	35.27	17.26	7.0 × 10${}^{-1}$	1.4 × 10${}^{-1}$	3.3 × 10${}^{-2}$
	D-ILJ-CBS-βFIT	35.18	17.15	6.8 × 10${}^{-1}$	1.4 × 10${}^{-1}$	3.2 × 10${}^{-2}$
	D-LJ-CBS	33.88	17.85	2.29	1.5 × 10${}^{-1}$	2.4 × 10${}^{-2}$
	Exp. [46]	32.0	–	–	–	–
	D-ILJ-β9	32.90	14.54	5.4 × 10${}^{-1}$	1.0 × 10${}^{-1}$	2.3 × 10${}^{-2}$
He-Kr	D-ILJ-CBS-β9	32.70	14.54	5.4 × 10${}^{-1}$	1.0 × 10${}^{-1}$	2.3 × 10${}^{-2}$
	D-ILJ-CBS-βFIT	32.57	14.37	5.1 × 10${}^{-1}$	1.0 × 10${}^{-1}$	2.3 × 10${}^{-2}$
	D-LJ-CBS	31.42	15.04	1.77	1.1 × 10${}^{-1}$	1.7 × 10${}^{-2}$
	Exp. [46]	29.1	–	–	–	–
	D-ILJ-β9	29.97	12.23	4.0 × 10${}^{-1}$	8.4 × 10${}^{-2}$	1.6 × 10${}^{-2}$
He-Xe	D-ILJ-CBS-β9	29.50	12.20	4.3 × 10${}^{-1}$	8.6 × 10${}^{-2}$	1.7 × 10${}^{-2}$
	D-ILJ-5z-βFIT	28.67	12.69	5.4 × 10${}^{-1}$	9.0 × 10${}^{-2}$	1.9 × 10${}^{-2}$
	D-LJ-CBS	28.34	12.62	1.39	9.4 × 10${}^{-2}$	1.2 × 10${}^{-2}$
	Exp.	–	–	–	–	–
He-Rn	D-ILJ-CBS-β9	28.52	11.45	3.9 × 10${}^{-1}$	7.8 × 10${}^{-2}$	1.5 × 10${}^{-2}$
	D-ILJ-CBS-βFIT	28.48	11.40	3.8 × 10${}^{-1}$	7.8 × 10${}^{-2}$	1.5 × 10${}^{-2}$
	D-LJ-CBS	27.40	11.85	1.27	8.5 × 10${}^{-1}$	1.1 × 10${}^{-1}$
	Exp. [46]	28.2	–	–	–	–
	D-ILJ-β9	27.07	4.47	6.5 × 10${}^{-2}$	1.3 × 10${}^{-2}$	1.0 × 10${}^{-3}$
	DVR-ILJ-CBS-β9	28.47	5.29	2.7 × 10${}^{-1}$	4.2 × 10${}^{-2}$	7.6 × 10${}^{-3}$
Ne-Ar	D-ILJ-CBS-β9	27.63	4.57	6.7 × 10${}^{-2}$	1.3 × 10${}^{-2}$	1.0 × 10${}^{-3}$
	DVR-ILJ-CBS-βFIT	28.74	5.43	2.9 × 10${}^{-1}$	4.3 × 10${}^{-2}$	7.6 × 10${}^{-3}$
	D-ILJ-CBS-βFIT	27.87	4.69	7.6 × 10${}^{-2}$	1.3 × 10${}^{-2}$	1.0 × 10${}^{-3}$
	DVR-LJ-CBS	26.74	4.92	2.8 × 10${}^{-1}$	5.0 × 10${}^{-2}$	4.1 × 10${}^{-3}$
	D-LJ-CBS	26.55	4.74	2.1 × 10${}^{-1}$	1.4 × 10${}^{-2}$	8.1 × 10${}^{-4}$
	Exp. [46]	26.2	–	–	–	–
	D-ILJ-β9	24.40	3.40	4.2 × 10${}^{-2}$	8.0 × 10${}^{-3}$	5.0 × 10${}^{-4}$
	DVR-ILJ-CBS-β9	25.12	3.72	1.3 × 10${}^{-1}$	2.8 × 10${}^{-2}$	3.6 × 10${}^{-3}$
Ne-Kr	D-ILJ-CBS-β9	24.77	3.42	4.2 × 10${}^{-2}$	8.3 × 10${}^{-3}$	5.7 × 10${}^{-4}$
	DVR-ILJ-CBS-βFIT	25.39	3.84	1.4 × 10${}^{-1}$	2.8 × 10${}^{-3}$	3.6 × 10${}^{-3}$
	D-ILJ-CBS-βFIT	25.02	3.52	4.8 × 10${}^{-2}$	8.0 × 10${}^{-3}$	5.0 × 10${}^{-4}$
	DVR-LJ-CBS	23.89	3.63	1.7 × 10${}^{-1}$	3.2 × 10${}^{-2}$	2.0 × 10${}^{-3}$
	D-LJ-CBS	23.80	3.54	1.3 × 10${}^{-1}$	9.1 × 10${}^{-3}$	4.2 × 10${}^{-4}$
	Exp. [46]	24.3	–	–	–	–
	D-ILJ-β9	22.58	2.81	3.0 × 10${}^{-2}$	6.0 × 10${}^{-3}$	3.0 × 10${}^{-4}$
	DVR-ILJ-CBS-β9	22.94	2.95	8.2 × 10${}^{-2}$	5.7 × 10${}^{-3}$	6.0 × 10${}^{-4}$
Ne-Xe	D-ILJ-CBS-β9	22.75	2.78	3.0 × 10${}^{-2}$	6.0 × 10${}^{-3}$	3.6 × 10${}^{-4}$
	DVR-ILJ-CBS-βFIT	23.36	3.11	9.7 × 10${}^{-2}$	5.8 × 10${}^{-3}$	6.1 × 10${}^{-4}$
	D-ILJ-CBS-βFIT	22.49	2.70	2.5 × 10${}^{-2}$	6.0 × 10${}^{-3}$	3.0 × 10${}^{-4}$
	DVR-LJ-CBS	21.91	2.94	1.2 × 10${}^{-1}$	6.5 × 10${}^{-3}$	3.4 × 10${}^{-4}$
	D-LJ-CBS	21.86	2.88	9.8 × 10${}^{-2}$	6.6 × 10${}^{-3}$	2.7 × 10${}^{-4}$

**Table 6 molecules-26-03906-t006:** Ne-Rn, Ar-Kr, Ar-Xe, Ar-Rn, Kr-Xe, Kr-Rn, and Xe-Rn rovibrational spectroscopic constants (RSC) given in cm${}^{-1}$.

Molecules	Methods	$\omega_{e}$	$\omega_{e}x_{e}$	$\omega_{e}y_{e}$	$\alpha_{e}$	$\gamma_{e}$
	Exp.	–	–	–	–	–
	D-ILJ-β9	–	–	–	–	–
	DVR-ILJ-CBS-β9	22.39	2.66	5.8 × 10${}^{-2}$	5.1 × 10${}^{-3}$	4.4 × 10${}^{-4}$
Ne-Rn	D-ILJ-CBS-β9	22.08	2.53	2.5 × 10${}^{-2}$	5.1 × 10${}^{-3}$	2.9 × 10${}^{-4}$
	DVR-ILJ-CBS-βFIT	22.27	2.61	5.4 × 10${}^{-2}$	5.1 × 10${}^{-3}$	4.4 × 10${}^{-4}$
	D-ILJ-CBS-βFIT	21.96	2.49	2.3 × 10${}^{-2}$	5.1 × 10${}^{-3}$	2.8 × 10${}^{-4}$
	DVR-LJ-CBS	21.42	2.68	9.7 × 10${}^{-2}$	5.8 × 10${}^{-3}$	2.5 × 10${}^{-4}$
	D-LJ-CBS	21.22	2.62	8.3 × 10${}^{-2}$	5.6 × 10${}^{-3}$	2.1 × 10${}^{-4}$
	Exp. [46]	27.9	–	–	–	–
	D-ILJ-β9	27.10	1.79	1.0 × 10${}^{-2}$	2.0 × 10${}^{-3}$	6.8 × 10${}^{-5}$
	DVR-ILJ-CBS-β9	27.23	1.81	1.6 × 10${}^{-2}$	2.0 × 10${}^{-3}$	9.0 × 10${}^{-5}$
Ar-Kr	D-ILJ-CBS-β9	27.22	1.80	1.0 × 10${}^{-2}$	2.1 × 10${}^{-3}$	6.9 × 10${}^{-5}$
	DVR-ILJ-CBS-βFIT	28.00	1.96	2.3 × 10${}^{-2}$	2.0 × 10${}^{-3}$	9.4 × 10${}^{-5}$
	D-ILJ-CBS-βFIT	27.44	1.84	1.0 × 10${}^{-2}$	2.0 × 10${}^{-3}$	7.0 × 10${}^{-5}$
	DVR-LJ-CBS	26.16	1.87	3.8 × 10${}^{-2}$	2.3 × 10${}^{-3}$	5.8 × 10${}^{-5}$
	D-LJ-CBS	26.15	1.86	3.4 × 10${}^{-2}$	2.3 × 10${}^{-3}$	5.1 × 10${}^{-5}$
	Exp. [46]	27.1	–	–	–	–
	D-ILJ-β9	25.71	1.43	7.0 × 10${}^{-3}$	1.0 × 10${}^{-3}$	3.9 × 10${}^{-5}$
	DVR-ILJ-CBS-β9	25.94	1.44	1.0 × 10${}^{-2}$	1.3 × 10${}^{-3}$	4.7 × 10${}^{-5}$
Ar-Xe	D-ILJ-CBS-β9	25.93	1.43	7.0 × 10${}^{-3}$	1.4 × 10${}^{-3}$	3.8 × 10${}^{-5}$
	DVR-ILJ-CBS-βFIT	25.42	1.36	7.5 × 10${}^{-3}$	1.0 × 10${}^{-3}$	4.6 × 10${}^{-5}$
	D-ILJ-CBS-βFIT	25.41	1.35	5.0 × 10${}^{-3}$	1.0 × 10${}^{-3}$	3.6 × 10${}^{-5}$
	DVR-LJ-CBS	24.92	1.48	2.4 × 10${}^{-2}$	1.5 × 10${}^{-3}$	3.1 × 10${}^{-5}$
	D-LJ-CBS	24.92	1.48	2.2 × 10${}^{-2}$	1.5 × 10${}^{-3}$	2.8 × 10${}^{-5}$
	Exp.	–	–	–	–	–
	D-ILJ-β9	–	–	–	–	–
	DVR-ILJ-CBS-β9	22.76	1.25	9.5 × 10${}^{-2}$	1.2 × 10${}^{-2}$	1.1 × 10${}^{-2}$
Ar-Rn	D-ILJ-CBS-β9	25.26	1.26	5.6 × 10${}^{-3}$	1.1 × 10${}^{-3}$	2.7 × 10${}^{-5}$
	DVR-ILJ-CBS-βFIT	22.64	1.23	8.7 × 10${}^{-3}$	1.1 × 10${}^{-3}$	3.8 × 10${}^{-5}$
	D-ILJ-CBS-βFIT	25.10	1.24	5.1 × 10${}^{-3}$	1.1 × 10${}^{-3}$	2.7 × 10${}^{-5}$
	DVR-LJ-CBS	21.71	1.25	2.1 × 10${}^{-2}$	1.3 × 10${}^{-3}$	2.4 × 10${}^{-5}$
	D-LJ-CBS	24.27	1.30	1.8 × 10${}^{-2}$	1.2 × 10${}^{-3}$	2.0 × 10${}^{-5}$
	Exp. [46]	22.7	–	–	–	–
	D-ILJ-β9	21.65	0.82	2.0 × 10${}^{-3}$	5.0 × 10${}^{-4}$	1.0 × 10${}^{-5}$
	DVR-ILJ-CBS-β9	21.58	0.81	3.4 × 10${}^{-3}$	1.9 × 10${}^{-3}$	4.2 × 10${}^{-5}$
Kr-Xe	D-ILJ-CBS-β9	21.58	0.81	2.7 × 10${}^{-3}$	5.4 × 10${}^{-4}$	1.0 × 10${}^{-5}$
	DVR-ILJ-CBS-βFIT	21.34	0.79	2.0 × 10${}^{-3}$	1.9 × 10${}^{-3}$	4.1 × 10${}^{-5}$
	D-ILJ-CBS-βFIT	21.34	0.78	2.0 × 10${}^{-3}$	5.0 × 10${}^{-4}$	9.9 × 10${}^{-6}$
	DVR-LJ-CBS	19.08	0.81	9.7 × 10${}^{-3}$	7.4 × 10${}^{-4}$	1.0 × 10${}^{-5}$
	D-LJ-CBS	20.73	0.84	8.8 × 10${}^{-3}$	5.9 × 10${}^{-4}$	7.6 × 10${}^{-6}$
	Exp.	–	–	–	–	–
	D-ILJ-β9	–	–	–	–	–
	DVR-ILJ-CBS-β9	19.09	0.66	2.6 × 10${}^{-3}$	3.9 × 10${}^{-4}$	7.8 × 10${}^{-6}$
Kr-Rn	D-ILJ-CBS-β9	20.42	0.66	1.9 × 10${}^{-3}$	3.8 × 10${}^{-4}$	6.2 × 10${}^{-6}$
	DVR-ILJ-CBS-βFIT	18.95	0.64	2.4 × 10${}^{-3}$	4.0 × 10${}^{-4}$	7.7 × 10${}^{-6}$
	D-ILJ-CBS-βFIT	20.25	0.65	1.7 × 10${}^{-3}$	3.8 × 10${}^{-4}$	6.1 × 10${}^{-6}$
	DVR-LJ-CBS	18.25	0.67	6.8 × 10${}^{-3}$	4.3 × 10${}^{-4}$	5.2 × 10${}^{-6}$
	D-LJ-CBS	19.62	0.69	6.3 × 10${}^{-3}$	4.2 × 10${}^{-4}$	4.7 × 10${}^{-6}$
	Exp.	–	–	–	–	–
	D-ILJ-β9	–	–	–	–	–
	DVR-ILJ-CBS-β9	17.88	0.45	1.2 × 10${}^{-3}$	2.0 × 10${}^{-4}$	2.8 × 10${}^{-5}$
Xe-Rn	D-ILJ-CBS-β9	17.98	0.45	1.0 × 10${}^{-3}$	2.0 × 10${}^{-4}$	2.5 × 10${}^{-6}$
	DVR-ILJ-CBS-βFIT	17.50	0.42	8.6 × 10${}^{-4}$	1.9 × 10${}^{-3}$	2.6 × 10${}^{-6}$
	D-ILJ-CBS-βFIT	17.81	0.43	8.7 × 10${}^{-4}$	2.0 × 10${}^{-4}$	2.4 × 10${}^{-6}$
	DVR-LJ-CBS	17.12	0.46	3.4 × 10${}^{-3}$	2.1 × 10${}^{-4}$	1.9 × 10${}^{-3}$
	D-LJ-CBS	18.06	0.47	3.2 × 10${}^{-3}$	2.1 × 10${}^{-4}$	1.7 × 10${}^{-6}$

## Data Availability

The data presented in this study are available in Appendix A.

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
