# Peer review of "A Spectroscopic Validation of the Improved Lennard–Jones Model"

_molecules, 2021, doi:10.3390/molecules26133906_

Round 1
Reviewer 1 Report
The paper is well written and results clearly presented. Some minor remarks:
In the methodology section, it is not stated which pseudopotential was used for the heavier elements. This is relevant for possible reproduction and extension of the work. It would also be worthy of discussion, whether the pseudopotential approximation provides adequate consideration of relativistic effects given the expected accuracy of the calculations.
I recommend mentioning in the abstract that the study touches the stability of the He-He dimer. This is a very long-standing debate and mentioning it there makes the paper easier to find in this context.
While the text is undoubdeldly understandable, the style and occasionally grammar can be improved. Enlisting help of a style editor (native speaker) could improve readability. Here are just a few of the problematic spots, there were several more:
Line 3 : "plain"? Perhaps "explain"?
Line 80: "no-covalent" -> "non-covalent"
Line 90: "plays"?
Line 143: "Table 22 of material information" Perhaps "supplementary information"?
Reviewer 2 Report
The manuscript by Mendes de Oliveira et al. describes the critical comparison between theoretical and experimental results on properties of noble gas dimers. The properties of dimers were described using improved Lennard-Jones potentials fitted to the results of extensive CCSD(T) calculations. Such studies are very valuable as they shed light on the functional form of non-covalent interactions and help the design of advanced interatomic potentials that are inspired by the Lennar-Jones method.
Since the discussion deals with highly precise computational and experimental data, the authors should also discuss the confidence intervals on the obtained fitted values of Re, De, and beta. For example, reliable confidence intervals can be estimated via bootstrap analysis.
In addition, the authors are suggested to discuss the importance of relativistic effects for the description of heavier atoms.
